# Self-Retrieval: End-to-End Information Retrieval with One Large Language Model

**Qiaoyu Tang**[1,2]*, **Jiawei Chen**[1,2]*, **Zhuoqun Li**[1,2], **Bowen Yu**[3], **Yaojie Lu**[1], **Cheng Fu**[3],
**Haiyang Yu**[3], **Hongyu Lin**[1†], **Fei Huang**[3], **Ben He**[1,2], **Xianpei Han**[1†], **Le Sun**[1], **Yongbin Li**[3†]
[1]Chinese Information Processing Laboratory, Institute of Software, Chinese Academy of Sciences
[2]University of Chinese Academy of Sciences
[3]Alibaba Group
{tangqiaoyu2020,jiawei2020,lizhuoqun2021}@iscas.ac.cn
{luyaojie,hongyu,xianpei,sunle}@iscas.ac.cn
{yubowen.ybw,fucheng.fuc,yifei.yhy,f.huang,shuide.lyb}@alibaba-inc.com
benhe@ucas.ac.cn

## Abstract

The rise of large language models (LLMs) has significantly transformed both the construction and application of information retrieval (IR) systems. However, current interactions between IR systems and LLMs remain limited, with LLMs merely serving as part of components within IR systems, and IR systems being constructed independently of LLMs. This separated architecture restricts knowledge sharing and deep collaboration between them. In this paper, we introduce *Self-Retrieval*, a novel end-to-end LLM-driven information retrieval architecture. Self-Retrieval unifies all essential IR functions within a single LLM, leveraging the inherent capabilities of LLMs throughout the IR process. Specifically, Self-Retrieval internalizes the retrieval corpus through self-supervised learning, transforms the retrieval process into sequential passage generation, and performs relevance assessment for reranking. Experimental results demonstrate that Self-Retrieval not only outperforms existing retrieval approaches by a significant margin, but also substantially enhances the performance of LLM-driven downstream applications like retrieval-augmented generation. [3]

## 1   Introduction

Recently, information retrieval (IR) systems and large language models (LLMs) have witnessed a growing synergy, with advancements in one field driving progress in the other [13, 56]. On one hand, IR systems have proven effective in augmenting LLMs and mitigating challenges such as hallucinations and outdated knowledge [22, 16]. By providing accurate, up-to-date external knowledge, IR systems significantly enhance the reliability and performance of LLMs. On the other hand, the powerful language understanding and generation capabilities of LLMs have been leveraged to enhance almost all components of traditional IR systems–indexing, retrieval [42, 9, 26], and reranking [58, 27, 40]. Through the integration of LLMs into the IR pipeline, these systems achieve substantially improved retrieval accuracy [57, 1].

However, current IR systems typically adopt a pipeline architecture where different components operate in isolation, limiting LLMs' role to specific components rather than leveraging their full

---

* Equally Contribution.
† Corresponding authors.
[3]The code of this work is available at `https://github.com/icip-cas/SelfRetrieval`.

38th Conference on Neural Information Processing Systems (NeurIPS 2024).

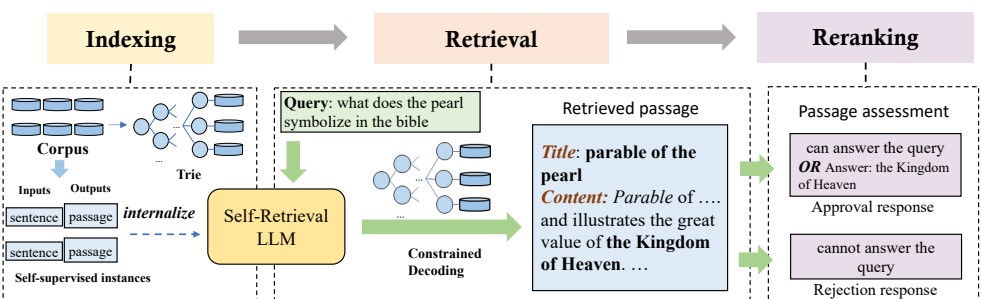

Figure 1: The Self-Retrieval framework consists of three key components: (1) corpus indexing through self-supervised learning, (2) passage generation via constrained decoding, (3) passage ranking using self-assessment scoring.

potential across the entire system. This fragmented approach creates several challenges: it hinders knowledge sharing between components, prevents deep integration of LLMs' diverse capabilities, and results in complex implementations with potentially sub-optimal performance. These limitations underscore the need for a more unified approach that fully integrates LLMs across all components of the IR system. Such an approach would not only maximize the utility of LLMs' capabilities but also simplify system implementation while potentially achieving better performance through enhanced component synergy.

In this paper, we introduce Self-Retrieval, an end-to-end information retrieval architecture driven entirely by one large language model. This integration is not trivial due to the inherent mismatch between information retrieval tasks and text generation, particularly in ensuring accurate document generation using language models. As illustrated in Figure 1, Self-Retrieval consolidates the separate components of an IR system - indexing, retrieval, and reranking - into the parameters of a single LLM. For indexing, the corpus is internalized into the LLM's parameters through self-supervised learning, enabling the model to encode and store corpus information within its internal representations. During retrieval, Self-Retrieval leverages its encoded knowledge of the corpus to semantically match the input query and directly generates the relevant documents as outputs. To ensure the generated documents exactly match those in the original corpus, we employ the constrained decoding algorithm [10, 8, 24] based on the trie of the corpus. For reranking, Self-Retrieval performs self-assessment on the retrieved documents to evaluate their relevance. The output score is used to rerank the retrieved passages. Moreover, for downstream tasks such as retrieval-augmented generation (RAG), Self-Retrieval integrates the reader component into the model, enabling direct answer generation following retrieval. Through this end-to-end approach, Self-Retrieval fully leverages LLMs' powerful capabilities in language understanding, matching, assessment, and generation to achieve unified information retrieval.

We evaluate Self-Retrieval on three representative retrieval benchmarks: NQ, TriviaQA, and MS MARCO. Experimental results demonstrate that Self-Retrieval substantially outperforms existing sparse retrieval, dense retrieval, and generative retrieval methods on both document-level and passage-level retrieval tasks. Furthermore, our experiments on retrieval-augmented generation tasks reveal that Self-Retrieval considerably enhances downstream performance. Additionally, larger LLMs lead to progressively better performance in Self-Retrieval, showing clear scaling benefits. These results demonstrate the effectiveness of Self-Retrieval across different retrieval tasks and application scenarios.

The potential impacts of this paper may include the following aspects. First, we introduce Self-Retrieval, an end-to-end architecture that consolidates the entire information retrieval system within a single large language model. This unified approach demonstrates substantial performance improvements over existing IR methods. Second, the corpus internalization and indexing mechanism of Self-Retrieval establishes a new paradigm to memorize, organize and retrieve the learned documents (at least part of them) during the pre-training phase, paving the way for more transparent and trustworthy text generation from LLMs. Third, as a LLM-driven retrieval system, Self-Retrieval offers inherent advantages in terms of compatibility, consistency, and interaction with LLMs' internal knowledge. Through experiments on RAG, we demonstrate how this natural compatibility leads to superior performance, suggesting broader potential for enhancing various LLM-based applications.

## 2 Related Work

**LLM for IR**   Recent studies have explored leveraging LLMs to enhance various components of IR systems, including query rewriting, retrieval, and reranking. For query rewriting, LLMs have been employed to generate pseudo-documents for query expansion [46] and to rewrite queries based on conversational context [15]. In the retrieval stage, researchers have explored augmenting data by generating pseudo-queries [6, 17] or relevance labels [25] using LLMs, as well as employing LLMs directly as generative retrievers [42, 5]. Regarding reranking, LLMs have been utilized in two ways: serving as rerankers directly [27, 40] and augmenting the reranking dataset [12]. While these methods have advanced specific components within the IR pipeline, Self-Retrieval distinguishes itself by presenting an end-to-end architecture driven entirely by a single LLM, eliminating the need for external components.

**Dense retrieval** Dense retrieval models retrieve information by matching dense vector representations of queries and documents [19]. In this paradigm, an encoder transforms both queries and documents into dense vectors, with relevance determined by their vector distance. Various strategies have been proposed to enhance dense retrievers, including designing loss functions [45], multi-vector [38], training with synthetic queries [33, 47], and leveraging large-scale query-document pairs [30, 50]. Recent work has also explored using large language models to generate dense vectors for both queries and documents [29]. However, the fundamental limitation of dense retrieval lies in its limited interaction with LLMs, as the compression of natural language into dense vectors inherently constrains the utilization of LLMs' sophisticated language understanding and semantic inference capabilities.

**Generative retrieval**   Generative retrieval methods leverage sequence-to-sequence language models to generate document identifiers for a given query [8, 42]. This paradigm is pioneered by GENRE [7], which introduces the concept of entity retrieval through constrained beam search generation of entity names. DSI [42] extends it to document retrieval by training T5 models to generate document-specific identifiers. The field has since evolved through various innovations, including query generation techniques [11, 59], sophisticated identifier design [48, 51], architectural improvements [5, 36], and continual learning strategies [20, 14].

Most relevant to our work, Yu et al.[52] proposed a "generate-then-read" approach, advocating for the use of LLMs to directly generate documents instead of relying on a retriever. UniGen [23] proposed a unified framework that integrates generative retrieval and question answering through a dual-decoder architecture. Compared to them, Self-Retrieval ensures accurate document generation through constrained decoding and accomplishes both retrieval and answer generation in one turn.

The main distinctions between Self-Retrieval and existing generative retrieval methods can be summarized as follows: (1) Self-Retrieval enables LLMs to directly generate document content rather than relying on other text or numeric identifiers. This approach aligns naturally with LLMs' pre-training objectives, preserves their inherent knowledge, and eliminates the need for complex identifier construction schemes. (2) Self-Retrieval further integrates components such as reranking and answer generation into the framework, further expanding its scope and enhancing the retrieval performance. These distinctions highlight that Self-Retrieval represents a more natural and effective approach for leveraging the capabilities of LLMs in information retrieval.

## 3 Self-Retrieval

In this section, we introduce our proposed Self-Retrieval. The overall architecture is illustrated in Figure 1. Different from traditional information retrieval systems that separate indexing, retrieval, and reranking components, Self-Retrieval integrates these functionalities directly into the parameters of a single large language model:

- **Indexing**: Self-Retrieval internalizes the entire corpus into its parameters through self-supervised learning, enabling the model to process passages internally without relying on external indices.
- **Retrieval**: Given an input query $q$, Self-Retrieval generates relevant passage $p$ using the knowledge embedded within its parameters, which is different from dense retrieval or generative retrieval that rely on embedding or document identifiers as proxies of passage.

- **Reranking**: After generating passage $p$, Self-Retrieval assesses its relevance to the query $q$ through self-assessment. The output logits provide the basis for reranking candidate passages.

Through this unified approach, Self-Retrieval enables a streamlined, end-to-end process that enhances the overall effectiveness of information retrieval. In the following sections, we detail each component of our method.

## 3.1 Indexing: Internalize the Corpus

Self-Retrieval integrates indexing into the LLM's parameters through self-supervised learning, enabling the model to internalize the entire corpus. Unlike generative retrieval methods that rely on complex document identifiers and identifier matching, Self-Retrieval employs a straightforward sentence-to-passage task to construct the index. Specifically, given a passage $p = \{s_1, s_2, ..., s_L\}$ consisting of $L$ sentences, each sentence $s_i$ is provided as input to the LLM with parameters $\theta$. The training objective is to generate the source passage $p$ in an auto-regressive way, represented as $P(p|s_i, \theta)$. This self-supervised indexing approach offers several advantages. First, it provides a simple yet effective method for corpus indexing. Second, it naturally frames the indexing process as a retrieval-like task, enabling the model to simultaneously internalize the corpus and develop retrieval capabilities using a consistent data format. Furthermore, this indexing technique closely aligns with the pre-training processes of language models, suggesting that our method could be considered as continued pre-training on the corpus. Through this process, the LLM learns to efficiently memorize and organize corpus information within its parameters.

## 3.2 Retrieval: Generate Relevant Passage through Constrained Decoding

Retrieval serves as a first-pass filter to collect passages related to the input query. In Self-Retrieval, we train the LLM to directly generate relevant passages in response to queries, eliminating the need for intermediaries such as embedding in dense retrieval or document identifier in generative retrieval. Specifically, given the query $q$ and corpus $\mathcal{D}$, Self-Retrieval first generates a potential document title $\hat{t}$ as global information, formulated as $P(\hat{t}|q; \theta)$. The model then generates a relevant passage, denoted as $P(\hat{p}|q, \hat{t}; \theta)$.

However, since LLMs are general-purpose pre-trained models rather than statistical frequency models, the generated passage $\hat{p}$ may not exactly match any passage in $\mathcal{D}$, making it challenging to locate the corresponding passages in the corpus. To address this challenge, we employ a trie-based constrained decoding algorithm [10, 8, 24]. This approach restricts generated tokens to a dynamically constrained vocabulary. We construct a prefix tree $\mathcal{T}$ from corpus $\mathcal{D}$, where each path from the root to a leaf node represents a unique passage in the corpus, and each node stores valid tokens for the next generation step. During inference, the vocabulary at each generation step is constrained by the valid continuations in the prefix tree. Due to the relatively short common prefixes among documents, the LLM terminates generation once it has produced sufficient tokens to uniquely identify the current document and concatenates the full document to the context. This results in document title and passage generation processes represented as $P(\hat{t}|q; \theta; \mathcal{T})$ and $P(\hat{p}|q, \hat{t}; \theta; \mathcal{T})$. This mechanism ensures that generated passages align with existing corpus content.

## 3.3 Reranking: Assess the Relevance

Reranking serves as a second-pass filter to precisely sort the retrieved passages based on the relevance to the query. We implement a self-assessment mechanism that leverages the Self-Retrieval model itself to evaluate the relevance of generated passages. Specifically, Self-Retrieval assesses the passage relevance by generating responses such as "can answer the query" for relevant passages and "cannot answer the query" for irrelevant ones. This self-assessment mechanism allows the model to generate passages and evaluate their relevance within a single inference turn.

During training, we utilize the gold passage from the supervision data as the positive instance, while sampling negative instances from both the same and different documents. This training strategy conditions the LLM to accurately discern and verify the relevance of its outputs, thereby enhancing its autonomous relevance assessment capabilities and improving the overall precision of the retrieval process.

During inference, the overall relevance score $\mathcal{S}$ is composed of the document title score $\mathcal{S}^T$ and the self-assessment score $\mathcal{S}^P$. Specifically, the document title score is derived from the title generation probability, while the self-assessment score is calculated based on the probability of the language model rejecting the passage. Formally, for a set of generated titles and passages $\{(t_1, p_1), (t_2, p_2), \ldots, (t_n, p_n)\}$, the title score for each $(t_i, p_i)$ is given by:

$$\mathcal{S}_i^T = \text{Softmax}(P(t_i|q; \theta)/\tau) \tag{1}$$

and the assessment score is:

$$\mathcal{S}_i^P = \text{Softmax}((1 - P(\text{rejection response}|q, t_i, p_i; \theta))/\delta) \tag{2}$$

where $\tau$ and $\delta$ are temperature parameters used to scale the logits. Based on preliminary experiments on the development set, we simply set $\tau = \delta = 0.4$ for the main passage retrieval experiments.

The final relevance score is computed as the product of these two components:

$$\mathcal{S} = \mathcal{S}^T \cdot \mathcal{S}^P \tag{3}$$

This combined score is then used to rerank the passage set, producing a more refined ordering based on relevance.

## 3.4 Training & Inference

**Training** Self-Retrieval unifies the three distinct tasks of information retrieval – indexing, retrieval, and reranking – into text generation tasks, trained using cross-entropy loss in an auto-regressive manner. Specifically, Self-Retrieval first internalizes the corpus into its parameters through self-supervised learning as introduced in Section 3.1. Subsequently, in addition to a portion of self-supervised instances, it incorporates two different types of data to build retrieval and reranking abilities:

- **Retrieval data:** Utilizes supervised query-passage pairs from the dataset, where the model learns to generate both document titles and passage content in response to input queries.
- **Reranking data:** Employs positive and negative examples to train the model in relevance assessment between queries and passages.

This auto-regressive training approach enables Self-Retrieval to integrate traditionally separate IR components into a unified language model, establishing an end-to-end IR system.

Furthermore, leveraging the universal language generation capabilities of LLMs, we can seamlessly integrate downstream task components, such as readers in RAG, into Self-Retrieval. This integration can be achieved by simply appending the golden answer after the assessment in Self-Retrieval. Consequently, the LLM can function as a comprehensive RAG system, effectively reducing the knowledge gap between IR system and reader modules.

**Inference** During inference, given an input query, Self-Retrieval aims to obtain the relevant passages that are sorted based on the relevance to query. Firstly, the model generates $i$ document titles through constrained beam search. Secondly, for each title, it generates $j$ passages using beam search. Finally, the resulting $i \times j$ passages are scored using the self-assessment mechanism and reranked to produce the final output.

# 4 Experimental Results

## 4.1 Experimental Setup

**Datasets and metrics** We conduct main experiments on Natural Questions (NQ) [21] and TriviaQA [18] datasets, both of which are widely used retrieval benchmarks based on Wikipedia. We use their versions from the KILT benchmark [34], which consolidates these datasets into a single pre-processed Wikipedia dump, facilitating easier evaluation. Since the KILT test set is not publicly accessible, we use the development set for testing and randomly sample 2,000 instances from the training set as our development set. For our experiments, we sample approximately 40K documents

| Model | Params | NQ | | | TriviaQA | | |
|---|---|---|---|---|---|---|---|
| | | H@1 | H@5 | M@5 | H@1 | H@5 | M@5 |
| *Sparse Retrieval* | | | | | | | |
| BM25 [37] | - | 14.54 | 32.71 | 21.13 | 20.09 | 42.73 | 28.35 |
| *Dense Retrieval* | | | | | | | |
| DPR [19] | 110M | 40.41 | 61.79 | 48.80 | 35.57 | 57.39 | 43.93 |
| DPR-FT [19] | 110M | 42.21 | 60.45 | 49.33 | 36.58 | 53.05 | 42.91 |
| BGE [50] | 335M | 36.30 | 66.95 | 48.05 | 46.97 | 70.14 | 55.95 |
| BGE-FT [50] | 335M | 53.42 | 80.15 | 63.99 | 52.70 | 75.22 | 61.65 |
| BGE-FT + BGE-Reranker-FT | 770M | 52.15 | 76.15 | 61.37 | 44.87 | 67.39 | 53.39 |
| GTR-XL [32] | 1.24B | 37.64 | 66.84 | 48.94 | 35.97 | 63.75 | 46.67 |
| GTR-XL + BGE-Reranker-FT | 1.57B | 57.50 | 78.92 | 66.06 | 58.56 | 77.65 | 66.22 |
| GTR-XXL [32] | 4.86B | 39.21 | 69.72 | 50.88 | 35.97 | 64.15 | 46.83 |
| text-embedding-ada-002 | - | 34.28 | 62.28 | 44.64 | 35.09 | 62.00 | 45.15 |
| GritLM [29] | 7.24B | 44.67 | 76.00 | 57.03 | 39.91 | 69.34 | 51.14 |
| GritLM + BGE-Reranker-FT | 7.57B | 57.57 | **81.35** | 66.98 | 58.60 | 80.54 | 67.21 |
| *Generative retrieval* | | | | | | | |
| DSI-XL [42] | 2.85B | 43.03 | 60.26 | 49.47 | 29.64 | 46.74 | 36.12 |
| DSI-XXL [42] | 11.3B | 43.81 | 60.45 | 50.20 | 30.55 | 46.67 | 36.56 |
| SEAL [5] | 406M | 36.79 | 61.35 | 45.88 | 36.88 | 61.66 | 46.29 |
| DSI-QG [59] | 2.85B | 34.88 | 56.60 | 43.33 | 29.15 | 45.53 | 35.20 |
| NCI + BGE-Reranker-FT | 1.07B | 50.86 | 70.27 | 58.53 | 28.42 | 42.18 | 33.62 |
| Self-Retrieval (StableLM) | 2.8B | 62.16* | 79.28 | 69.45* | 58.69* | 78.39* | 66.72* |
| Self-Retrieval (Llama 2) | 6.74B | **63.44*** | 79.29 | **70.00*** | **59.94*** | **81.06*** | **68.74*** |

Table 1: The experimental results of passage retrieval on NQ and TriviaQA test set. * indicates statistically significant improvements (p < 0.01) over state-of-the-art retrieval baselines.

from Wikipedia for each dataset. Each document is segmented into passages of maximum 200 words, yielding approximately 1 million passages in total. The detailed statistics of the datasets are presented in Appendix A. We use passage-level Hits@{1, 5} and Mean Reciprocal Rank (MRR)@5 as evaluation metrics.

To comprehensively compare with other generative information retrieval methods, we also conduct experiments on document retrieval. Following NCI [49], we conduct experiments on NQ320K and utilize Recall@{1, 10} and MRR@100 as the evaluation metrics. To evaluate the model's robustness in non-Wikipedia scenarios where high-quality text and titles are not available, we conduct experiments on a subset of MS MARCO [3] following the experimental setup of Ultron [55]. The performance was measured using Recall@{1,5} and MRR@10.

**Implementation details** In this study, we employ StableLM-3B [44] and Llama2-7B [43] as passage retrieval backbones. For document retrieval, we employ StableLM-1.6B [4] for NQ320K and StableLM-3B for MS MARCO. We train the models using ZeRO stage-2 optimization on 8 NVIDIA A100 (80 GB) GPUs with the AdamW optimizer, a batch size of 16 per GPU, and BFloat16 precision. The models are trained for 3 epochs with a learning rate of 2e-5. During inference, we use beam search to generate 5 titles and 10 passages for each title, with hyperparameters $\tau$ and $\delta$ set to 0.4 across all models and datasets.

**Baselines** We evaluate Self-Retrieval models for both passage retrieval and document retrieval, comparing them with sparse, dense, and generative retrieval baselines. The **sparse retrieval** baselines are: BM25 [37] and DocT5Query [28]. The **dense retrieval** baselines include: DPR [19], Sentence-T5 [31], GTR [32], BGE [50], text-embedding-ada-002 [30], GritLM [29], and their fine-tuned variants, DPR-FT and BGE-FT. The **generative retrieval** baselines comprise: DSI [42], DSI-QG [59], NCI [49], Ultron [55], DynamicRetriever [54], GenRet [39], and SEAL [5]. Additionally, to ensure a comprehensive comparison, we also evaluate combinations of strong retrieval baselines with various rerankers, including BGE-Reranker, BGE-Reranker-FT, and RankGPT [41]. In the passage retrieval task, we use the official pre-trained models for all non-fine-tuned dense retrieval baselines. For fine-tuned dense models and generative models, we use their official implementations to replicate the experiments on our dataset. In the document retrieval task, we report the baseline

performances from their original paper. For comprehensive details about these baselines, please refer to Appendix B.

## 4.2 Main Results

**Passage retrieval** In Table 1, we compare the performance of Self-Retrieval with various baselines on the NQ and TriviaQA datasets. Self-Retrieval 3B outperforms both strong pre-trained dense retrieval models, such as BGE and GritLM 7B, and other generative retrieval methods. Specifically, Self-Retrieval 3B achieves improvements of 5.46 and 5.07 in MRR@5 over the fine-tuned BGE on NQ and TriviaQA datasets, respectively.

Our results indicate that other generative retrieval baselines exhibit suboptimal performance on passage retrieval. Even the largest DSI-XXL model only achieves an MRR@5 of 50.20 on NQ, significantly lagging behind dense retrieval methods such as GritLM, which achieves an MRR@5 of 57.03. In contrast, our Self-Retrieval model demonstrates strong performance in passage retrieval, achieving an MRR@5 of 69.45, significantly outperforming all other generative methods.

We further compare Self-Retrieval with conventional 2-stage retriever-reranker pipeline. Representative results are shown in Table 1, while the complete experimental results are provided in Appendix D. Notably, even strong retrieval baselines (BGE-FT, GTR-XL, GritLM, and DSI-XL) enhanced with powerful rerankers (such as BGE-Reranker-FT) still fall short of Self-Retrieval's performance, highlighting the advantages of unifying multiple retrieval processes into a single framework rather than treating them as separate components.

These findings underscore the efficacy of Self-Retrieval in harnessing the memory, generation, and ranking capabilities of LLMs, thereby excelling in passage retrieval tasks where other generative baselines struggle.

| Method | R@1 | R@10 | M@100 |
|---|---|---|---|
| *Sparse Retrieval* | | | |
| BM25 [37] | 29.7 | 60.3 | 40.2 |
| DocT5Query [28] | 38.0 | 69.3 | 48.9 |
| *Dense Retrieval* | | | |
| DPR [19] | 50.2 | 77.7 | 59.9 |
| Sentence-T5 [31] | 53.6 | 83.0 | 64.1 |
| GTR-Base [32] | 56.0 | 84.4 | 66.2 |
| *Generative Retrieval* | | | |
| DSI [42] | 55.2 | 67.4 | 59.6 |
| SEAL [5] | 59.9 | 81.2 | 67.7 |
| DSI-QG [59] | 63.1 | 80.7 | 69.5 |
| NCI [49] | 66.4 | 85.7 | 73.6 |
| GenRet [39] | 68.1 | 88.8 | 75.9 |
| Self-Retrieval | **73.3** | **92.6** | **80.7** |

Table 2: The experimental result of document retrieval on NQ320K.

| Method | R@1 | R@5 | M@10 |
|---|---|---|---|
| *Sparse Retrieval* | | | |
| BM25 [37] | 18.9 | 42.8 | 29.2 |
| DocT5Query [28] | 23.3 | 49.4 | 34.8 |
| *Dense Retrieval* | | | |
| DPR [19] | 29.1 | 62.8 | 43.4 |
| Sentence-T5 [31] | 27.3 | 58.9 | 40.7 |
| *Generative Retrieval* | | | |
| DSI-Atomic [42] | 32.5 | 63.0 | 44.3 |
| DynamicRetriever [54] | 29.0 | 64.2 | 42.5 |
| Ultron-URL [55] | 29.6 | 56.4 | 40.0 |
| Ultron-PQ [55] | 31.6 | 64.0 | 45.3 |
| Ultron-Atomic [55] | 32.8 | 64.9 | 46.9 |
| GenRet [39] | **47.9** | - | **58.1** |
| Self-Retrieval | 47.8 | 69.9 | 57.2 |

Table 3: The experimental result of document retrieval on MS MARCO.

**Document retrieval** We present the document retrieval results on NQ320K dataset in Table 2. Self-Retrieval outperforms all other generative retrieval methods and dense retrieval baselines across all three metrics. Compared to GenRet, the previously strongest generative retrieval method, Self-Retrieval improves Hits@1 by 5.2, Hits@10 by 3.8, and MRR@100 by 4.8 points. Notably, while other methods commonly employ query generation to augment their training data, Self-Retrieval achieves these results using only the original training set.

To evaluate the effectiveness of Self-Retrieval in non-Wikipedia scenarios, we extend our experiments to MS MARCO. To address the absence of document titles in MS MARCO, we employ Llama2 to automatically generate titles. As shown in Table 3, Self-Retrieval achieves comparable performance to the SOTA model GenRet, while significantly outperforming other baselines. These results demonstrate its adaptability and robustness in non-Wikipedia and title-lacking contexts.

**Ablation study** To study the effect of each component, we conduct ablation study on both NQ and TriviaQA. Results are presented in Table 4. All components prove crucial for Self-Retrieval's

| Method | NQ | | | TriviaQA | | |
|---|---|---|---|---|---|---|
| | H@1 | H@5 | M@5 | H@1 | H@5 | M@5 |
| Self-Retrieval (base) | 62.16 | 79.28 | 69.45 | 58.69 | 78.39 | 66.72 |
| w/o indexing | 53.05 | 67.16 | 58.95 | 54.45 | 70.64 | 60.98 |
| w/o title | 47.81 | 60.90 | 52.81 | 52.32 | 67.91 | 58.48 |
| w/o self-assessment | 54.80 | 75.21 | 62.77 | 46.67 | 70.79 | 55.92 |

Table 4: Ablation study on NQ and TriviaQA.

performance, with each ablation resulting in substantial performance degradation. Specifically, removing the indexing mechanism restricts the model to internalizing only the documents encountered during training, leading to poor performance on unseen passages. Without titles, we directly generate passages with constrained decoding. The absence of document titles significantly degrades performance, as titles provide critical global information that guides the LLM in generating relevant content. Furthermore, removing the self-assessment mechanism leads to a significant decrease in both datasets. Without self-assessment, the model cannot effectively evaluate and refine its initial retrieved passages, leading to less accurate document rankings. This degradation directly impacts downstream applications such as RAG, where precise passage ranking is crucial for generating high-quality responses. These ablation results show that each component of Self-Retrieval addresses a specific challenge in the retrieval process, contributing to its overall effectiveness.

## 4.3 Performance on Retrieval-Augmented Generation

| | NQ | | TriviaQA | |
|---|---|---|---|---|
| | 10K | 40K | 10K | 40K |
| BGE-FT + StableLM-FT | 43.18 | 41.24 | 56.79 | 58.15 |
| Self-Retrieval 3B | 44.62 | 46.11 | 64.03 | 62.69 |
| BGE-FT + Llama2-FT | 49.10 | 49.24 | 61.79 | 61.72 |
| Self-Retrieval 7B | 53.26 | 52.98 | 72.14 | 70.40 |

Table 5: The performance on retrieval-augmented generation. For baseline, we use BGE-FT as the retriever and a fine-tuned LLM as reader. Results are reported using Exact Match (EM) scores.

The end-to-end architecture of Self-Retrieval seamlessly integrates retrieval and answer generation into a single inference process. To evaluate its effectiveness in RAG, we compare Self-Retrieval models with a strong baseline that combines BGE-FT for retrieval and fine-tuned versions of StableLM-3B and LLaMA2-7B as readers. We conduct experiments on subsets of NQ and TriviaQA using 10K and 40K documents for each dataset. We utilize the top-1 retrieved passage as the context and measure performance using the Exact Match (EM) metric. As shown in Table 5, Self-Retrieval significantly outperforms the baseline on both datasets across different model scales. Unlike other RAG pipelines that separate retrieval and generation, Self-Retrieval integrates the entire process within the LLM framework, enabling more accurate and coherent responses through end-to-end modeling.

## 4.4 Detailed Analysis

**Scaling model capacity** To explore the impact of model scale on retrieval performance, we evaluate Self-Retrieval with various backbone models of different sizes, including StableLM (1.6B, 3B) [4, 44], Llama2 (7B, 13B) [43], and Qwen-1.5 (4B, 7B, 14B) [2]. Figure 2 presents the results on NQ, showing that Self-Retrieval's retrieval performance benefits from the general capabilities of larger language models. For models within the same series, as the model size increases, we observe consistent improvements in both Hits@1 and Hits@5, indicating strong scaling properties of the Self-Retrieval architecture.

**Scaling corpus size** Recent studies [35, 53] have demonstrated that generative retrieval methods such as DSI or NCI experience more significant performance degradation compared to dense retrieval methods when scaled to larger corpora. To explore the impact of corpus size on Self-Retrieval,

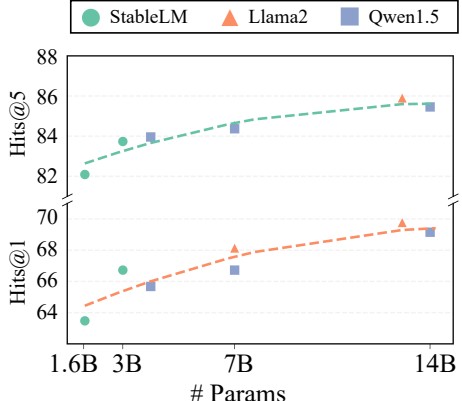

Figure 2: Impact of model capacity on Self-Retrieval performance.

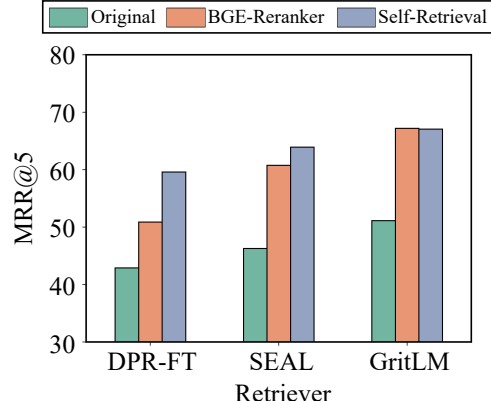

Figure 3: Reranking performance comparison when processing top-100 passages.

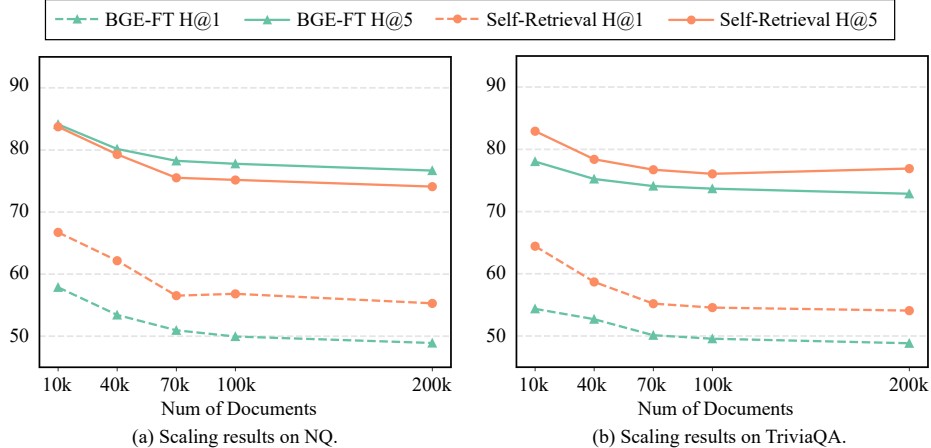

(a) Scaling results on NQ.
(b) Scaling results on TriviaQA.

Figure 4: Scalability analysis of retrieval performance for Self-Retrieval and BGE-FT across varying corpus sizes.

we expand our experiments from 10K to 200K documents, scaling the number of passages from 290K to 3M. Figure 4 illustrates the performance trends of BGE-FT and our Self-Retrieval 3B model on the NQ and TriviaQA datasets with increasing corpus sizes. While both models show performance decrease with larger corpus sizes, Self-Retrieval maintains a degradation rate comparable to BGE-FT. As the number of documents continues to increase, the degradation rate gradually diminishes, demonstrating Self-Retrieval's potential scalability to larger document collections. This observation indicates that Self-Retrieval effectively addresses some of the inherent limitations of generative retrieval approaches in large-scale scenarios.

**Analysis on reranking** In this part, we conduct an in-depth analysis of the reranking performance of Self-Retrieval reranker module in comparison with the fine-tuned BGE-Reranker. We employ DPR-FT, SEAL and GritLM to retrieve 100 passages on TriviaQA, followed by reranking the retrieved results using both approaches. We evaluate performance using MRR@5 as the metric. The experimental results are presented in Figure 3. The results reveal two key findings: (1) Reranking plays a crucial role in information retrieval systems, significantly enhancing the ranking performance across all models. (2) The Self-Retrieval reranker consistently outperforms the fine-tuned BGE Reranker in most scenarios, demonstrating its robustness and effectiveness. These findings demonstrate that Self-Retrieval performs effectively both as a complete IR system and as a reranker component.

In Appendix C, we conduct additional experiments with a chunk size of 100 words, demonstrating Self-Retrieval's adaptability to different text segmentation strategies. In Appendix E, we further discuss Self-Retrieval's computational efficiency.

## 5   Conclusion

In this paper, we propose Self-Retrieval, an end-to-end LLM-driven information retrieval architecture that unifies indexing, retrieval, and reranking in a single LLM. This approach enables the LLM to internalize the corpus, generate relevant content, and perform self-assessment within a unified framework. Unlike previous works that incorporate LLMs into individual IR components, Self-Retrieval provides a unified framework for the entire IR procedure, facilitating knowledge sharing and deep collaboration among different components. Experimental results demonstrate that Self-Retrieval achieves strong performance across various retrieval benchmarks and application scenarios. In future work, we plan to extend our method to further enhance the reliability and trustworthiness of LLM generation.

## Limitations

While our experiments demonstrate the effectiveness of Self-Retrieval, several limitations need to be addressed in future work. Our current evaluation is limited to 200K Wikipedia documents and 3M passages, and testing on larger and noisier text collections is needed. As an LLM-driven system, Self-Retrieval has lower retrieval efficiency compared to sparse or dense retrieval methods, which may limit its applications to specialized knowledge systems. Furthermore, enabling incremental learning and dynamic corpus expansion remains an important direction for future research.

## Acknowledge

We sincerely thank the reviewers for their insightful comments and valuable suggestions. We are grateful to Le Yu and Xinyu Lu for their helpful feedback on the paper writing. This work was supported by the Natural Science Foundation of China (No. 62122077, 62272439), Beijing Municipal Science and Technology Project (Nos. Z231100010323002), the Basic Research Program of ISCAS (ISCAS-JCZD-202303), and CAS Project for Young Scientists in Basic Research (Grant No.YSBR-040).

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

# A Dataset Statistics

Table 6 presents the statistics of the NQ and TriviaQA datasets used in our experiments.

| Dataset | Natural Questions | | TriviaQA | |
|---|---|---|---|---|
| | 10K | 40K | 10K | 40K |
| # doc | 10,000 | 37,202 | 10,000 | 38,399 |
| # psg | 291,506 | 979,804 | 390,586 | 1,193,047 |
| # train | 32,163 | 72.716 | 29,038 | 51,166 |
| # dev | 2,000 | 2,000 | 2,000 | 2,000 |
| # test | 2,837 | 2,837 | 5,355 | 5,355 |

Table 6: Statistics of the experimental datasets. #doc/#psg denotes number of documents/passages; #train/#dev/#test denotes size of training/development/test set. Training instances without query-document pairs are removed.

# B Baselines

The sparse retrieval baselines are as follows:

- **BM25** [37] is a classical sparse retrieval algorithm based on probabilistic relevance framework and term frequency statistics.
- **DocT5Query** [28] expands documents by generating potential queries using a fine-tuned T5 model.

The dense retrieval baselines are as follows:

- **DPR** [19] is a dual-encoder model trained with in-batch negative sampling. We fine-tune DPR on our training datasets to obtain **DPR-FT**, following the official implementation and hyperparameter settings.
- **BGE** [50] is a state-of-the-art universal embedding model trained on approximately 200 million text pairs using contrastive learning. We employ the bge-large-en-v1.5 variant and fine-tune it on our training datasets to obtain **BGE-FT**. The fine-tuning process uses a learning rate of 1e-5, batch size of 128, and runs for 10 epochs.
- **Sentence-T5** [31] employs a dual-encoder T5 architecture to generate semantic embeddings through contrastive learning for efficient retrieval.
- **GTR-XL** [32] is a dense retrieval model based on Sentence-T5, pre-trained on billions of question-answer pairs.
- **Text-embedding-ada-002** is a powerful embedding model developed by OpenAI, accessible through their API service.
- **GritLM** [29] is built upon the Mistral 7B language model and optimized using both embedding and generation objectives.

The generative retrieval baselines are as follows:

- **DSI** [42] is a sequence-to-sequence model that directly maps queries to document identifiers.
- **DSI-QG** [59] enhances the DSI framework by incorporating a doc2query model for dataset augmentation.
- **SEAL** [5] utilizes n-gram as the document identifiers and constrains the generation process using FM-index.
- **NCI+BGE-Reranker-FT**. NCI [49] employs a sequence-to-sequence architecture with a prefix-aware weight-adaptive decoder. We train the model using T5-Large for document-level retrieval following the official implementation. To obtain passage-level results, we further incorporate a fine-tuned BGE reranker (bge-reranker-large).

- **Ultron** [55] represents documents using three types of identifiers (URL, PQ, Atomic) and trains the model through a progressive three-stage pipeline.
- **DynamicRetriever** [54] parameterizes traditional static indices by embedding both token-level and document-level corpus information into a pre-trained model for dynamic document identifier generation.
- **GenRet** [39] employs discrete auto-encoding with progressive training and clustering techniques to learn semantic document identifiers for generative retrieval.

## C Ablation on Chunk Size

To investigate the potential impact of chunk size, we conduct additional experiments comparing Self-Retrieval with strong baselines on the NQ dataset using a chunk size of 100 words, complementing our main experiments where chunk size is set to 200. The experimental results are presented in Table 7. It demonstrates that Self-Retrieval significantly outperforms the baselines with both chunk sizes settings, further validating the effectiveness of our proposed method.

| Model | Params | Hits@1 | Hits@5 | MRR@5 |
|---|---|---|---|---|
| BGE-FT | 335M | 40.79 | 58.92 | 47.76 |
| GritLM | 7B | 30.95 | 51.36 | 38.77 |
| Self-Retrieval (StableLM) | 3B | **58.43** | **77.76** | **66.18** |

Table 7: Retrieval performance with chunk length of 100 words.

## D Full Comparison with Retriever-Reranker Pipeline

| | NQ | | | TriviaQA | | |
|---|---|---|---|---|---|---|
| | H@1 | H@5 | M@5 | H@1 | H@5 | M@5 |
| BGE-FT | 53.42 | 80.15 | 63.99 | 52.70 | 75.22 | 61.65 |
| BGE-FT + BGE-Reranker | 21.91 | 54.58 | 33.33 | 45.36 | 72.16 | 55.78 |
| BGE-FT + BGE-Reranker-FT | 52.15 | 76.15 | 61.37 | 44.87 | 67.39 | 53.39 |
| BGE-FT + RankGPT | 44.21 | 73.68 | 55.51 | 48.00 | 72.00 | 57.33 |
| GTR-XL | 37.64 | 66.84 | 48.94 | 35.97 | 63.75 | 46.67 |
| GTR-XL + BGE-Reranker | 26.39 | 59.96 | 38.50 | 42.41 | 68.42 | 52.51 |
| GTR-XL + BGE-Reranker-FT | 57.50 | 78.92 | 66.06 | 58.56 | 77.65 | 66.22 |
| GTR-XL + RankGPT | 42.11 | 68.42 | 52.30 | 47.00 | 66.00 | 54.95 |
| GritLM | 44.67 | 76.00 | 57.03 | 39.91 | 69.34 | 51.14 |
| GritLM + BGE-Reranker | 30.06 | 65.87 | 43.20 | 43.64 | 70.87 | 54.23 |
| GritLM + BGE-Reranker-FT | 57.57 | **81.35** | 66.98 | 58.60 | 80.54 | 67.21 |
| GritLM + RankGPT | 37.89 | 70.53 | 51.19 | 44.00 | 66.00 | 52.70 |
| DSI-XL | 43.03 | 60.26 | 49.47 | 29.64 | 46.74 | 36.12 |
| DSI-XL + BGE-Reranker | 34.39 | 64.26 | 45.74 | 37.85 | 52.57 | 43.49 |
| DSI-XL + BGE-Reranker-FT | 50.02 | 68.60 | 57.43 | 36.49 | 52.40 | 42.36 |
| DSI-XL + RankGPT | 49.47 | 73.68 | 59.25 | 39.00 | 52.00 | 44.75 |
| Self-Retrieval (StableLM) | 62.16 | 79.28 | 69.45 | 58.69 | 78.39 | 66.72 |
| Self-Retrieval (Llama 2) | **63.44** | 79.29 | **70.00** | **59.94** | **81.06** | **68.74** |

Table 8: Comparison between Self-Retrieval and traditional two-stage retriever-reranker pipelines.

We comprehensively evaluate Self-Retrieval against various two-stage retriever-reranker pipelines. Specifically, we construct these pipelines using state-of-the-art retrievers (BGE, GTR, GritLM, and DSI-XL) combined with three different reranking approaches: BGE reranker, fine-tuned BGE reranker, and RankGPT. As shown in Table 8, Self-Retrieval achieves superior performance compared to most retriever-reranker combinations, demonstrating the effectiveness of our end-to-end approach over traditional pipeline methods.

# E    Efficiency Analysis

We conduct efficiency analysis on NQ dataset using an NVIDIA A100-80G GPU. Results in Table 9 illustrate that, while Self-Retrieval requires slightly higher computational resources than DSI, it provides notable performance benefits. Notably, Self-Retrieval with a beam size of 10 achieves significantly higher H@5 scores compared to DSI-XL with a beam size of 100, enabling a flexible trade-off between retrieval quality and computational efficiency. When compared to SEAL, which also employs natural language decoding, Self-Retrieval demonstrates more efficient memory usage (30MB vs 444MB) by utilizing a lightweight trie structure instead of SEAL's resource-intensive FM-Index post-processing mechanism. Furthermore, the efficiency of Self-Retrieval stands to benefit from ongoing developments in optimization techniques (e.g., quantization and attention acceleration) and hardware advancements.

| Model Name | Memory | Beam Size | Latency (s) | Hits@5 |
|---|---|---|---|---|
| SEAL | 444MB | 10 | 1.18 | 61.91 |
|  |  | 100 | 5.92 | 59.57 |
| DSI-XL | 0 | 10 | 0.23 | 60.21 |
|  |  | 100 | 0.45 | 60.21 |
| Self-Retrieval | 30MB | 10 | 1.44 | 76.17 |
|  |  | 100 | 6.06 | 81.49 |

Table 9: Efficiency analysis.

