# OpenReview forum: "Self-Retrieval: End-to-End Information Retrieval with One Large Language Model"
_NeurIPS.cc/2024/Conference — NeurIPS 2024 poster_

### Official Review · Reviewer_uLuX · 2024-07-08

**Soundness:** 3
**Presentation:** 3
**Contribution:** 3
**Rating:** 5
**Confidence:** 4

**Summary:**

This paper introduces Self-Retrieval, an end-to-end IR system driven entirely by a single LLM. This model integrates all essential IR functions—indexing, retrieval, and reranking—into the LLM's architecture. By internalizing the retrieval corpus through self-supervised learning, the model transforms the retrieval process into a sequence of passage generation tasks and conducts self-assessment for reranking. The authors provide experimental evidence showing that Self-Retrieval outperforms traditional sparse, dense, and generative retrieval methods on benchmark datasets like NQ and TriviaQA.

**Strengths:**

1. The integration of all IR functions into a single LLM is a novel contribution that leverages the inherent capabilities of LLMs across the full spectrum of IR tasks, offering a streamlined and potentially more effective approach.
2. The concept of Self-Retrieval is introduced clearly, making it accessible to readers. The detailed explanation of how the LLM handles indexing, retrieval, and reranking provides a good understanding of the system's operation.
3. The paper presents good experimental results that demonstrate significant improvements over existing retrieval methods.

**Weaknesses:**

1. My major concern is that the experimental settings are inconsistent with existing work, making the results unconvincing. Specifically,
  - Most existing studies conducted experiments on NQ@320k datasets, but the main experiments of this paper are conducted on NQ@40k datasets. It is important to explain the reason of this setting.
  - According to the statistics in Table 2, each document in the dataset is split into 26~29 passages, which indicates that the passages are quite short. It is known that retrieving short passages are easier for generative retrieval methods.
  - It is necessary to explain why different models are selected for NQ@40k, TQA@40k, and NQ@320k experiments.
  - It is better to include more experimental results on the full KILT datasets as many existing studies for a fair comparison.
2. In Section 3.4, the three tasks are learned in a "1+2" manner. Please add more explanations on this design and provide experimental evaluation on other possible strategies (e.g., training the three tasks jointly)
3. Ensuring consistent use of key terms throughout the paper would improve its readability and professionalism. For example, all "large language models" should be written in "LLMs".

**Questions:**

Please see my concerns in weaknesses.

---

> ### Author Rebuttal · Authors · 2024-08-07
>
> Thanks for your time and insightful comment. We would like to address your questions in turn.
> ### About NQ@40k in our experiments
> Currently, the most common and widely used retrieval method is dense retrieval, which primarily focuses on passage-level retrieval ([2, 3]). Consequently, we constructed NQ@40k to specifically evaluate performance at the passage level. Additionally, we observed that most other generative retrieval methods struggle with passage-level retrieval, whereas self-retrieval successfully overcomes this limitation.
> ### The length of passages
> In common settings for passage-level retrieval, documents are often segmented into chunks of 100 or 200 tokens ([1, 2]). Therefore, we believe that using 200 tokens as the passage length for our main experiment is reasonable. Additionally, we have reported the performance for the more commonly used 100-token segments in the Appendix. It should be noted that shorter passages may not necessarily lead to better performance, according to [1].
>
> [1] Multi-passage BERT: A Globally Normalized BERT Model for Open-domain Question Answering.
>
> [2] Dense Passage Retrieval for Open-Domain Question Answering.
>
> [3] C-pack: Packaged resources 535 to advance general chinese embedding.
>
> ### The differences in baselins of NQ@40k and NQ@320k
> 1. NQ@40k is a passage-level retrieval setting,  following the setting of previous dense retrieval works. We reproduced the latest dense retrieval methods like BGE and GirtLM. Since some previous generative retrieval works have not been open-sourced, we have only reproduced and compared some key generative retrieval baselines.
> 2. NQ@320K is a document-level retrieval setting, following the setting of previous generative retrieval works to ensure comparability with previous works.
> Furthermore, thank you for your reminder. In the subsequent versions, we will add the performance of the latest dense retrieval methods, such as BGE and GritLM, on NQ@320k for comparison.
>
> ### About  full KILT datasets
> Due to time and computational resource limitations, we were unable to complete training and testing on the entire Wikipedia corpus from KILT during the rebuttal period. Our analysis of scaling corpus size in Figure 3 of the main paper shows that Self-Retrieval maintains a relatively parallel performance relationship with BGE-FT as the document size increases. This trend indicates that Self-Retrieval has substantial scaling potential. Furthermore, due to the scaling characteristics inherent to LLMs, we also believe that their capacity to handle documents has a very high upper limit. However, due to the entire Wikipedia corpus comprsing around 5.9M documents, it is challenging to complete training within the effective rebuttal period. We appreciate your insightful comment and will add more scaling experiments in the future.
> ### Experiments on different training strategies
> Different training strategies indeed have a certain impact on Self-Retrieval's performance. Our choice of the "1+2" training strategy was confirmed through early preliminary experiments.  To compare different strategies, we conducted experiments on NQ40k. We found that jointly training on all three datasets yields overall performance (79.26 Hits@5) comparable to the "1+2" strategy(79.28 Hits@5). However, the "1+2" training method allows multiple different training sets with the same corpus to share a single indexing process. This significantly reduces the overall training time.
> ### About wrting
> Thank you for your suggestion. We will refine it in the next version.

---

> ### Author Response · Authors · 2024-08-14
>
> Dear Reviewer uLuX,
>
> As the discussion deadline is nearing, we wanted to gently follow up on our recent submission. We've carefully addressed your feedback and incorporated it into our revised manuscript. We provided clarifications on our choice of NQ@40K for alignment with standard passage-level experiments in dense retrieval, and explained why the "1+2" strategy is preferable to joint training in our context.
> We hope our rebuttal have addressed your concerns regarding our experimental and training settings.
>
> Your feedback is highly valuable to us. We welcome any additional feedback or questions you may have about our revisions.
>
> Thank you for your time and expertise.

---

### Official Review · Reviewer_iXLD · 2024-07-10

**Soundness:** 3
**Presentation:** 3
**Contribution:** 3
**Rating:** 6
**Confidence:** 4

**Summary:**

This paper proposes Self-Retrieval, an LM that retrieve, rerank passages, and generate answers using a single model. For the retrieval task, it adopts generative retrieval and directly generates passage text. For reranking, it utilizes the generation probability as the relevance score. For answer generation, it generates answers based on the top-1 passage. Evaluation shows that Self-Retrieval outperforms previous dense and generative retrievers in retrieval tasks and achieves better EM scores in answer generation tasks.

**Strengths:**

1. Self-Retrieval consolidates the multi-step RAG pipeline with a single model. The concept is novel.

2. Self-Retrieval achieves the best performance for both passage retrieval and answer generation tasks.

3. Self-Retrieval shows promising results when the corpus size scales to 3 million.

**Weaknesses:**

1. Generating passage content is time-consuming. This paper could analyze the latency of Self-Retrieval and compare it to other alternatives, such as generating spans (ref SEAL) or keywords (ref Term-Set Generation).

2. The proposed model includes an in-domain fine-tuned reranker, while the baseline BGE-FT + reader does not have a reranking stage. This may make the comparison unfair since reranking can significantly improve RAG results.

3. The evaluation are all based on Wikipedia. It is unclear whether the model can perform well on a corpus that is not as well-structured as Wikipedia.

4. The different steps in Self-Retrieval are independently optimized, and the `knowledge sharing and deep collaboration` effects of this consolidated model have not been validated.

**Questions:**

1. Compare the efficiency and effectiveness of using shared models versus separate models for the three steps in Self-Retrieval?

**Limitations:**

The limitations are adequately discussed.

---

> ### Author Rebuttal · Authors · 2024-08-07
>
> Thank you for your suggestions on our work. Here is our response to your concerns.
> ### Analyze the latency
> We compared the efficiency of Self-Retrieval with SEAL in the General Response Table 1. Our generation process has the following characteristics:
> 1. Early Stop Mechanism: Our method includes an 'early stop' mechanism where the Self-Retrieval model ceases generation once the LLM has produced sufficient content to determine the current passage. As shown in Attachment Figure 1, the average decision length of the trie is approximately 13 tokens, which is similar to the decision length in Term-Set Generation. Consequently, the total number of tokens generated remains relatively small.
> 2. No Additional Post-Processing: Unlike SEAL, our Self-Retrieval method does not require extra time for post-processing or retrieving passages. This means that even when using a larger LLM, the overall efficiency remains comparable to that of SEAL.
>
> ### About reranker in RAG
> We have augmented our experiments with the incorporation of a BGE reranker in the  RAG process (BGE-FT + BGE reranker-FT + StableLM-FT). EM scores on the NQ and TriviaQA datasets were recorded as 41.98 and 50.16, respectively. As can be seen from the results, there was no observed improvement over the BGE baseline that does not employ a reranker. This outcome is likely due to the BGE's substantial proficiency developed during the training phase. Consequently, as shown in Attachment Table 1, additional training of the reranker does not bring extra benefit.
> ### Experiments on non-wikipedia datasets
> Please refer to the General Response.
>
> ### Shared Model v.s. Separated Models
> We separately trained individual retrieval and ranking models, achieving Hit@1 and Hit@5 scores of 60.56 (compared to 62.16 with joint training) and 78.21 (compared to 79.28 with joint training) on NQ@40K. These results show a slight decline compared to joint training in Self-retrieval. This indicates that Self-Retrieval can synergize the retrieval and ranking tasks during training to achieve better performance. Additionally, using shared models can reduce deployment costs.

---

> > ### Comment · Reviewer_iXLD · 2024-08-11
> >
> > Thank you for your detailed reply, which addresses many of my concerns.
> > ﻿
> > Regarding the comparison with DSI and SEAL, does the latency / H@5 only account for the retrieval step of Self-Retrieval, or does it also include the re-ranking step? If it includes the re-ranking step, can we still perform early stopping during passage generation, as in Eq 2, the re-ranking score seems to rely on the full passage content.
> >
> > Thanks.

---

> ### Author Response · Authors · 2024-08-12
>
> Thanks for your time and valuable comment.
> ### Efficiency Analysis Details:
>
> |Model Name|Memory|Beam Size|Latency(s)|H@5|
> |---|---|---|---|---|
> |SEAL|444MB|10|1.18|61.91|
> |||100|5.92|59.57|
> |DSI-XL|0|10|0.23|60.21|
> |||100|0.45|60.21|
> |Self-Retrieval|30MB|10|1.44|76.17|
> |||100|6.06|81.49|
> |Self-Retrieval w/o re-ranking|30MB|10|1.36|74.04|
> |||100|4.57|72.97|
>
> The latency/H@5 results in Table 1 of Global Response include the re-ranking step. We have updated the table to include results without re-ranking as well. These results demonstrate that without re-ranking, Self-Retrieval further improves efficiency while maintaining competitive performance.
> ### Early Stopping Mechanism Details:
> When the Early Stopping Mechanism is triggered, we have typically decoded only a few tokens. Nevertheless, these tokens are sufficient to identify a unique passage. In the prefix tree, each leaf node corresponds to a specific passage ID, allowing us to map the decoded prefix directly to a passage ID. Subsequently, we extract the full passage content from the corpus and append it to the previously decoded content for re-ranking.

---

### Official Review · Reviewer_g3kT · 2024-07-14

**Soundness:** 3
**Presentation:** 4
**Contribution:** 3
**Rating:** 6
**Confidence:** 5

**Summary:**

This paper introduces Self-Retrieval, a new generative retrieval architecture.  Self-Retrieval first memorizes the corpus into LLM's parametric knowledge using self-supervised training. Given a query, it generates the target document with constrained decoding, then re-assess document by decoding if the document can answer the query.  On subset of NQ and TriviaQA, this approach significantly outperforms existing dual encoders and generative retrieval models.

**Strengths:**

- Intuitive architecture for using LLM for retrieval. The paper presents a self-supervise object to help model memorize the corpus. Then it integrate retrieval and reranking into a single constrained decoding process. The "reranking" process can be viewed as self-critique / chain-of-thoughts, and may potentially unlock deeper integration between retrieval and reasoning.
- Strong quality improvements. The paper reports substantial improvements over previous dual-encoder approaches and generative retrieval models.
- Ablations shows that the model scales well with model size. Previous dense retrievers often plateau due to the bottleneck layer; the scaling curve of this new architecture is promising.

**Weaknesses:**

- Experiment only used wikipedia-based datasets. However, wikipedia is heavily used in pretraining, so it is unclear if the proposed approach can let model sufficiently memorize other datasets.
- More importantly, the method relies on generating the passage title. Unlike wikipedia, many retrieval datasets do not have high-quality, natural language passage titles. It is unclear how the method works on those datasets.  It would be nice to test retrieval benchmarks like MS MARCO or BEIR/MTEB.
- Missing dense retrieval + cross-attention reranking baselines. Such 2-staged pipeline is standard in IR. Since the proposed method's reranking stage essentially uses cross attention to judge the query and the retrieved candidate passage, the computational cost of the reranking stage is similar to that of a separate cross-attention reranker. It is fair and necessary to compare it with commonly-used rerankers such as MonoT5, RankT5, or BGE reranker.  The ablation in Table 3 seems to show that the proposed method's retrieval-alone performance is stronger than most retrieval baselines, but the paper can be more convincing if having e2e comparison to other 2-stage retrieval pipelines like BGE + BGE reranker or GTR + RankT5.
- Lacking efficiency discussion.

**Questions:**

- Can the method scale up to the full NQ/TriviaQA? If so, would be nice to see the performance on these more standard setups. If not, what is the main bottleneck for scaling up?
- How many candidates were considered in the reranking stage?

---

> ### Author Rebuttal · Authors · 2024-08-07
>
> Thank you for your constructive comments. We appreciate your feedback and will address each of your points in turn.
>
> ### Experiments on non-wikipedia datasets
> Please refer to the General Response.
> ### Experiments on untitled documents
> With untitled documents or titles of poor quality, Self-Retrieval can use the generated model from other models. We conducted experiments on MS MARCO. Specifically, we prompted LLaMA3 8B in a zero-shot manner to generate appropriate titles for the documents missing the title. We tested the settings following GenRet on MS MARCO. The experimental results, shown in the General Response Table 2, indicate that titles generated by the LLM can effectively meet Self-Retrieval needs.
>
> ### About retrieval+reranker baselines
> Thank you for your feedback. We conducted experiments using a 2-staged retriever+reranker approach. Specifically, we chose strong retrieval baselines such as BGE, GTR, GritLM, and DSI-XL as the foundation, and then applied three different rerankers: BGE reranker, BGE reranker FT, and RankGPT. (We did not use RankT5 because the relevant code or model has not been open-sourced).
> As shown in Attachment Table 1, our Self-Retrieval approach outperforms most retriever+reranker combinations, demonstrating the effectiveness of our method.
>
> ### Efficiency discussion
> Please refer to the General Response.
>
> ### Scaling up to full NQ/TriviaQA
> In Figure 3 of our paper, we demonstrate the performance of Self-Retrieval and BGE as the corpus size increases. Our experiments included a maximum of approximately 200k documents and 3M passages. We are currently conducting larger-scale NQ/TriviaQA experiments to further validate the scaling effects of our model. However, due to the full NQ/TriviaQA datasets involving the entire Wikipedia corpus (approximately 21M passages for the DPR version and 5.9M documents for the KILT version), it is challenging to complete training within the effective rebuttal period. We appreciate your insightful comment and plan to include more scaling experiments in the future.
>
> ### Number of candidates in the reranking stage
> In the reranking stage, we use 50 passages as candidates.

---

> > ### Comment · Reviewer_g3kT · 2024-08-13
> >
> > Thank you for the response and the additional experimental results! The results addressed one of my major concerns. I'm still not quite clear about the scaling / efficiency of the proposed method. I'll keep my score unchanged.

---

> ### Author Response · Authors · 2024-08-14
>
> We sincerely appreciate your time and valuable feedback. We are pleased that our additional experimental results have addressed one of your primary concerns.
>
> Regarding the scaling capabilities of Self-Retrieval:
>
> 1. As shown in Figure 3 of the main paper, our current experiments encompass a substantial dataset of approximately 200k documents and 3 million passages, demonstrating Self-Retrieval's capability with large-scale collections.
> 2. It's worth noting that in generative information retrieval research, using partial document collections is common due to training efficiency constraints. For instance, Ultron and GenRet utilized subsets of 300k documents from MS MARCO, while UniGen employed a subset of approximately 100k passages from the NQ dataset for evaluation. In contrast, Self-Retrieval conducted main experiments on 1 million passages of NQ and further expanded it to 3 million on the scaling experiment.
> 3. We are actively conducting experiments on the full NQ dataset. However, due to time and computational resource limitations during the rebuttal period, we were unable to complete these extensive experiments for this response.
>
> Concerning efficiency, as mentioned in our Global Response, Self-Retrieval achieves comparable efficiency to SEAL while maintaining high performance. Moreover, Self-Retrieval requires only a single LLM to manage both retrieval and downstream LLM tasks. This unified architecture enhances deployment flexibility and potentially reduces computational overhead.
>
> We hope these points can partially alleviate your concern about the scaling and efficiency aspects of our method. We remain committed to further large-scale experiments and are open to addressing any additional questions or concerns you may have.

---

### Official Review · Reviewer_xB99 · 2024-07-23

**Soundness:** 3
**Presentation:** 3
**Contribution:** 3
**Rating:** 6
**Confidence:** 4

**Summary:**

The paper proposes an approach of self-retrieval, which uses the probability of generation of the passage as the ranking criterion. To limit the generation to the existing passages, a trie-structure is used, forcing the generation to produce the existing passages.
The experiments compared the method with several existing approaches, including sparse retrieval, dense retrieval and generation-based retrieval. The proposed method outperforms the others.

**Strengths:**

The idea of relying on the generation of an existing passage for ranking is interesting. The use of trie structure to constrain the generation to the existing documents is also nice. The proposed method thus has some novelty compared to the literature.
The experimental results are very good, showing improved performance on document retrieval and QA.

**Weaknesses:**

A key idea is the use of trie for passage generation. This is described briefly. More details should be presented. If a whole passage should be generated using trie, then the depth of trie structure will be very large (equivalent to the length of the passage). What about the storage cost of trie? What is the time efficiency?

**Questions:**

Have you evaluated the space and time complexity of the approach, and compare it to the existing methods?
How do you deal with long document (in particular for the trie structure)?

**Limitations:**

yes

---

> ### Author Rebuttal · Authors · 2024-08-07
>
> Thanks for your helpful comments! We are very glad to address your concerns one by one.
>
> ### Details of Trie
> The trie is pre-built based on the corpus before retrieval. Most documents/passages only share a small common prefix. The LLM stops generating once it has produced enough tokens to determine the current document. As shown in Figure 1 of the attachment file, the average decision length of the trie is around 13 tokens. For most documents, the retrieval process can decide on the current document after generating less than 20 tokens, at which point we "early stop" and manually append the document to the context.
>
> In this case, the depth of the trie depends more on the number of documents/passages rather than the length of each document/passage. Therefore, even when dealing with long documents, the common prefix length between documents may not necessarily be long, allowing Self-Retrieval to handle long documents effectively.
> We will include these details in the next version.
> ### Space and Time Complexity
> Please refer to the General Response.

---

### Author Rebuttal · Authors · 2024-08-07

We sincerely thank all reviewers for their insightful comments and valuable suggestions. In our responses, we provided details of the Trie structure to Reviewer xB99, additional experiments on untitled documents and retrieval + reranker baselines to Reviewer g3kT, along with clarifications regarding scaling up to full NQ/TriviaQA and the number of candidates in the reranking stage. We also addressed the latency analysis and the reranker in RAG for Reviewer iXLD and clarified our dataset selection and training strategies for Reviewer uLuX. Here, we would like to provide a unified response to the common concerns raised by the reviewers regarding space and time complexity, as well as experiments on non-Wikipedia datasets as follows:

### Space and Time Complexity

Table 1
|Model Name|Memory|Beam Size|Latency(s)|H@5|
|---|---|---|---|---|
|SEAL|444MB|10|1.18|61.91|
|||100|5.92|59.57|
|DSI-XL|0|10|0.23|60.21|
|||100|0.45|60.21|
|Self-Retrieval|30MB|10|1.44|76.17|
|||100|6.06|81.49|

The table compares the time and space efficiency of Self-Retrieval with other typical generative retrieval methods on NQ40K.
1. Compared to DSI:
   - Although Self-Retrieval has some disadvantages in terms of time and space efficiency, it offers a significant performance advantage. Even with a beam size of 10, its performance far exceeds that of DSI with a beam size of 100. This allows for a trade-off between performance and efficiency when using Self-Retrieval.
   - Additionally, with the advent of software and hardware accelerations for large models, the performance of Self-Retrieval is expected to improve further.

2. Compared to SEAL:
   - Both Self-Retrieval and SEAL use natural language decoding, but SEAL employs an FM-Index, which results in extensive post-processing after generating the natural language, severely impacting efficiency.
   - Self-Retrieval uses a smaller additional storage structure compared to SEAL. The trie, with early stopping, has an average length of 13, resulting in significantly less additional storage than the FM-Index used by SEAL.

### Experiments on non-wikipedia datasets

Table 2
|Method|R@1|R@5|MRR@10|
|---|---|---|---|
|BM25|18.9|42.8|29.2|
|DocT5Query|23.3|49.4|34.8|
|Sentence-T5|27.3|58.9|40.7|
|DPR|29.1|62.8|43.4|
|DSI-Atomic|32.5|63.0|44.3|
|DynamicRetriever|29.0|64.2|42.5|
|Ultron-Atomic|32.8|64.9|46.9|
|GenRet|47.9|-|58.1|
|Self-Retrieval|47.8|69.9|57.2|

This is an excellent suggestion. We conducted experiments on the MS MARCO dataset to verify the effectiveness of Self-Retrieval on non-Wikipedia documents. The MS MARCO dataset consists of crawled web pages that differ from the structured content and titles found in Wikipedia. It contains various complex, repetitive, and lower-quality information and titles. Following Ultron and GenRet, we trained and tested on a subset of MS MARCO and used DocT5Query to generate additional training data. We utilized StableLM-3B as the backbone model and followed the training and inference methods described in Section 3.4.

The experimental results, as shown in the Table, indicate that on the MS MARCO dataset, Self-Retrieval outperforms most generative baselines such as Ultron and DSI, as well as dense baselines like DPR. This demonstrates that Self-Retrieval can still perform well on non-Wikipedia documents.

---

### Decision · Program_Chairs · 2024-09-25

**Decision:**

Accept (poster)

**Comment:**

This paper introduces Self-Retrieval, an approach that uses LLMs for end-to-end information retrieval tasks. The proposed approach integrates indexing, retrieval, and reranking in a single model and utilizes a novel strategy for passage generation and self-assessment.

The reviewers found the paper to be well-written, the proposed approach to be highly novel, and the experimental results to be convincing and strong.

The reviewers did identify a number of weaknesses, as well, including deviations from standard evaluation practices (e.g., passage-based setup vs. document based one, KILT setup, etc.), the desire to see more detailed ablations, and questions around the computational efficiency of the approach.

Overall, this is a very interesting and novel paper that will very likely open up opportunities for future research along similar directions. While the paper isn't perfect, its strengths outweigh its weaknesses. The authors are encouraged to carefully consider the reviewer feedback when preparing an updated version of the paper.